# Gastric cancer incidence, mortality and burden in adolescents and young adults: a time-trend analysis and comparison among China, South Korea, Japan and the USA

Si Lin Wu,[1] Yao Zhang,[2] Yi Fu,[2] Jian Li  ,[2] Ji Sheng Wang[3]

[1]School of Pharmacy, Southwest Medical University, Luzhou, Sichuan, China
[2]Department of General Surgery, The Third Hospital of Mianyang, Sichuan Mental Health Center, Mianyang, Sichuan, China
[3]Department of Pharmacy, The Third Hospital of Mianyang, Sichuan Mental Health Center, Mianyang, Sichuan, China

**Correspondence to**
Dr Jian Li; 654747973@qq.com

## ABSTRACT

**Objectives** To evaluate and compare the burden of gastric cancer in adolescents and young adults (GCAYA) among China, South Korea, Japan and the USA, four countries with similar or different rates of gastric cancer (GC) incidence, development levels and cancer control strategies.

**Design** This population-based observational study collected the epidemiological data of GCAYA from the Global Burden of Diseases Study 2019. The trend magnitude and directions over time for incidence and mortality of GCAYA were analysed and compared among four countries.

**Main outcomes and measures** Outcomes included new cases, deaths, mortality-to-incidence ratios (MIRs), disability-adjusted life years, and their age-standardised rates and estimated annual percentage changes (AAPCs).

**Results** There were 49 008 new cases and 27 895 deaths from GCAYA in 2019, nearly half of which occurred in China. The AAPCs for the age-standardised incidence and mortality rate were 0.3 (−0.1 to 0.7), −3.6 (−3.7 to −3.4), −3.2 (−3.8 to −2.6), −0.1 (−0.6 to 0.5) and −2.0 (−2.3 to −1.6), −5.6 (−6.2 to −5.0), −4.4 (−4.7 to −4.1), −0.7 (−1.0 to −0.3) in China, South Korea, Japan and the USA, respectively. The incidence rate for females in the USA rose by 0.4% annually. GC ranks fifth, first, fourth and ninth in China, South Korea, Japan and the USA regarding burdens caused by cancer in adolescents and young adults. The MIRs declined constantly in South Korea and China, and the MIR in the USA became the highest in 2019.

**Conclusions** Although not covered by prevention and screening programmes, variations in disease burden and time trends may reflect variations in risk factors, cancer control strategies and treatment accessibility of GC among the four countries. Investigating the reasons behind the varying disease burden and changing trends of GCAYA across countries will inform recommendations for prevention measures and timely diagnosis specific to this underserved population to further decrease the GC burden.

## STRENGTHS AND LIMITATIONS OF THIS STUDY

⇒ We provided a comprehensive description of variations in the incidence and mortality of gastric cancer in adolescents and young adults (GCAYA) among China, South Korea, Japan and the USA.
⇒ Our study uses the average annual percentage change and the annual percentage change to quantify and compare secular trends in the incidence and mortality of GCAYA.
⇒ This study analyses the mortality-to-incidence ratios of GCAYA and their changing trends among China, South Korea, Japan and the USA.
⇒ We were unable to analyse cardia and non-cardia gastric cancer separately, two subtypes that have different risk factors and temporal incidence trends.
⇒ The incidence and mortality were low and volatile, especially in the USA, which means that even the smallest change could lead to a significant analytical outcome.

## INTRODUCTION

Gastric cancer (GC) has long been a major disease burden caused by neoplasms worldwide.[1] Recent evidence suggests that the incidence and mortality of GC in the general population has fallen substantially,[2] primarily resulting from the prevention and nationwide screening programmes.[3 4] On the contrary, a possible rising incidence of early-onset GC has been reported in the USA.[5 6] However, the incidence and disease burden caused by GC in the USA were relatively smaller than those caused by other cancer types. In addition, there are no nationwide screening programmes for GC in the USA. In Japan and South Korea, and in recent years in China, population screening has been performed widely, although none of them covered people younger than 40 years old.[7 8] The trends of GC incidence in youth populations have also been reported in Asian countries. In Japan, no marked changes in the incidence of GC were noted for individuals aged 30–39.[9] The results from the South Korean study showed a falling trend in the 20–39 age group.[10] However, the end time of

the analysis period in these studies was 10–30 years ago or before the implementation of nationwide screening programmes. Hence, trends in recent years and whether prevention and screening programmes also influence the incidence and mortality of GC in adolescents and young adults (GCAYA), are unknown.

Given that adolescents and young adults (AYAs) represent the main proportion of people who contribute substantially to the economy and have an important role in caring for their families, GCAYA carries a disproportionate burden than GC among older patients due to its greater impact on life expectancy.[11 12] Variations in cancer incidence among different populations may reflect differences in the prevalence of risk factors and screening strategies. Variations in mortality reflect variations not only in incidence but also in case fatality, which can be affected by differences in early diagnosis and accessibility to treatment.[13] Therefore, we conducted a comprehensive analysis of the rates and trends of incidence, mortality and disability-adjusted life years (DALYs) for GCAYA in China, South Korea, Japan and the USA, four countries with similar or different rates of GC incidence, development levels and cancer control strategies. We collected all data from the Global Burden of Diseases, Injuries, and Risk Factors Study 2019 (GBD 2019). By investigating the differences in the burden and changing trends of GCAYA among the four countries, we hope that our findings can serve as a reference for the establishment of GCAYA control measures and help to reduce the disease burden caused by this neglected cancer type.

## METHODS

### Study population and data sources

In this study, the research subjects were AYAs diagnosed with GC. AYA were defined as individuals aged 15–39. We obtained all data analysed in this study from GBD 2019, which aims to analyse health trends over time, compare variability among countries and help establish disease control strategies globally.[14] We collected data from the Global Health Data Exchange (GHDx) (http://ghdx.healthdata.org/) via the freely available GBD Results Tools repository. The search parameters were "stomach cancer" for cause; "incidence, deaths, DALYs" for measurements; "China, Republic of Korea, Japan, United States of America" for location; "1990–2019" for years; "number and rate" for metrics; "male, female and both" for sex; and "15 to 39 years and corresponding 5-year bands" for age. We followed the Guidelines for Accurate and Transparent Health Estimates Reporting guidelines for cross-sectional studies.[15]

### Definitions

The definition of GCAYA is not always consistent across studies, yet most authors adopted 40 years as the upper limit to categorise patients as having early-onset GC.[12] Therefore, in this study, we defined GCAYA as patients diagnosed between the ages of 15 and 39 years. The

rationale for using this age range relates to biological and physiological maturity and relative stability; these individuals have not yet experienced the effects of hormonal and immune response decline or chronic medical conditions that can influence oncological decision-making as it would in the care of older patients.[16] The DALY is a summary measure that quantifies the overall burden of disease, which represents the sum of years of life lost due to premature death and years lived with disability. One DALY can be regarded as the loss of 1 year in full health.

### Patient and public involvement

Patients and/or the public were not involved in the design, conduct, reporting or dissemination plans of this research.

### Statistical analysis

Detailed estimation methods for incidence, mortality and DALYs have been reported in previous studies by GBD Collaborators.[14 17] We computed the age-standardised incidence rate (ASIR) and age-standardised mortality rate (ASMR) using the crude rates of 5-year bands from 15 to 39, and the GBD 2019 standard population via the direct method, expressed as the rate per 100 000 person-years. We analysed incidence, mortality and DALYs descriptively by gender, country and year, and we calculated the change rates between 1990 and 2019. We also calculated the mortality-to-incidence ratio (MIR)—which has previously been employed as a proxy for the 5-year survival rate across different neoplasias—as the ratio of death counts to new cases.[18–20] We plotted the temporal trends of these measures from 1990 to 2019. To compare the changing trends of GCAYA among the four countries, we used Joinpoint software (V.4.9.0.0) to determine the average annual percentage change (AAPC) and the annual percentage change (APC) for each period, with a maximum of two joinpoints using a generalised linear regression model for the natural logarithm of the ASIR and ASMR. We established the statistical significance of the variation trend by their 95% CIs. We considered AAPCs or APCs with a 95% CI of >0 to represent a significant rising trend, while we deemed those with a 95% CI of <0 to represent a significant falling trend; otherwise, they represented a stable ASIR or ASMR.[21 22]

## RESULTS

### New cases of GCAYA and its change rates between 2019 and 1990

In 2019, there were an estimated 1 269 806 new GC cases globally, 49 008 (3.86%) of which were diagnosed between 15 and 39 years old. China accounted for 42.55% (20 855) of GCAYA cases. As shown in table 1, in South Korea and Japan, new cases of GCAYA were common in females, while in China and the USA, GCAYA was much more frequently diagnosed in males. Compared with that in 1990, the new cases of GCAYA declined by 58.51% in South Korea and 70.99% in Japan, and the degrees of

**Table 1** New cases and deaths of gastric cancer in adolescents and young adults, and percentage changes from 1990 to 2019 in China, South Korea, Japan and the USA

| | | New cases | | | Deaths | | |
|---|---|---|---|---|---|---|---|
| | | 1990 | 2019 | 1990–2019 change (%) | 1990 | 2019 | 1990–2019 change (%) |
| China | Both | 18 123 | 20 855 | 15.07 | 13 929 | 8 462 | −39.25 |
| | Male | 9 803 | 14 005 | 42.86 | 7 464 | 5 508 | −26.21 |
| | Female | 8 320 | 6 851 | −17.66 | 6 465 | 2 955 | −54.29 |
| Korea | Both | 1 921 | 797 | −58.51 | 1 254 | 237 | −81.10 |
| | Male | 904 | 352 | −61.06 | 571 | 101 | −82.31 |
| | Female | 1 017 | 445 | −56.24 | 682 | 136 | −80.06 |
| Japan | Both | 3 258 | 945 | −70.99 | 1 239 | 273 | −77.97 |
| | Male | 1 626 | 462 | −71.59 | 538 | 131 | −75.65 |
| | Female | 1 632 | 483 | −70.40 | 700 | 142 | −79.71 |
| USA | Both | 772 | 812 | 5.18 | 400 | 343 | −14.25 |
| | Male | 450 | 441 | −0.02 | 223 | 174 | −21.97 |
| | Female | 322 | 370 | 14.91 | 177 | 169 | −4.52 |

reduction were similar in males and females. However, new cases in China and the USA have risen by 15.07% and 5.18%, respectively. The increased number of new cases in China contributed to male cases, while in the USA it contributed to female cases.

### GCAYA-related deaths and their change rates between 2019 and 1990

In 2019, the number of deaths caused by GC was 957 185 worldwide, and GCAYA accounted for only 2.91% (27 895). China contributed to 13 929 (49.93%) of the deaths caused by GCAYA. The sex distribution was similar to that of new cases; females predominated in China and the USA, while males predominated in South Korea and Japan. In contrast to new cases, the number of deaths between 2019 and 1990 declined in all four countries. The most obvious changes occurred in South Korea, reaching more than 80% for both sexes. The lowest decline was among females in the USA, which was only 4.52% (table 1).

### Age-standardised rates and time trends of GCAYA incidence

As shown in table 2 and figure 1, for both sexes, the ASIRs of GCAYA in 2019 in China, South Korea, Japan and the USA were 3.71, 3.99, 2.55 and 0.71 per 100 000 person-years, respectively. Consistent with the sex variations in new cases, the ASIRs were higher for females than for males in Japan and South Korea, while the opposite was true in the USA and China. The variability of ASIR was also found through time-trend analysis among the four countries. Only in Japan did the ASIR exhibit a constant declining trend, with AAPC values of −3.6 (−3.7 to −3.4) for both sexes. In South Korea, there was a decreasing trend for both males (AAPC −3.4, 95% CI −4.5 to −2.2) and females (AAPC −2.7, 95% CI −2.9 to −2.5), although the ASIR in males tended to remain stable after 2016. The shifting characteristics of ASIRs in China are much more complex. The changing trends were not significant from

1990 to 2019, with an AAPC of 0.3 (−0.1 to 0.7), resulting from a considerably falling trend from 2004 to 2014 (APC −1.6, 95% CI −2.3 to −0.8) but a significantly rising trend from 2014 to 2019 (APC 2.4, 95% CI 0.4 to 4.4). The ASIR of GCAYA in the USA was low and remained relatively stable in males; however, the ASIR in females rose by 0.4% annually from 1990 to 2019.

### Age-standardised rates and time trends of GCAYA mortality

In 2019, the ASMRs of GCAYA in China, South Korea, Japan and the USA were 1.50 (1.27 to 1.75), 1.18 (0.94 to 1.47), 0.73 (0.68 to 0.78) and 0.30 (0.27 to 0.33), respectively. A decreasing trend of ASMR was observed from 1990 to 2019 in all four countries, and the annual decline rates were 2.0%, 5.6%, 4.4% and 0.7% in China, South Korea, Japan and the USA, respectively. The decrease started at approximately 2000 in China for females; before that time, it had been rising for 10 years (APC 0.8, 95% CI 0.0 to 1.6). For males in China, among the total falling trend, there was a stable period (1997–2003). The downward trend continued in China and the USA until 2019, but stabilised in South Korea and Japan from 2016 (table 3; figure 2).

### DALYs caused by GCAYA and its change rates between 2019 and 1990

The GBD 2019 estimated that GCAYA resulted in 475 977, 13 267, 15 367 and 19 233 DALYs in China, South Korea, Japan and the USA, respectively. The corresponding age-standardised DALY rates (ASDR) were 84.68, 66.67, 41.67 and 16.85 per 100 000 person-years. Similar to incidence and mortality, female predominance was noted in South Korea and Japan, while male predominance was witnessed in China and the USA. Between 1990 and 2019, the ASDR declined in all four countries. The proportions of reduction were 38.97%, 81.44%, 77.71% and 13.98% in China, South Korea, Japan and the USA, respectively (online supplemental table 1). Compared with other

**Table 2** The temporal trend in the incidence rate of gastric cancer in adolescents and young adults from 1990 to 2019 in China, South Korea, Japan and the USA

| Country | Sex | ASIR (per 100 000) | | Trends 1 | | Trends 2 | | Trends 3 | | 1990–2019 |
|---|---|---|---|---|---|---|---|---|---|---|
| | | 1990 | 2019 | Years | APC (95% CI) | Years | APC (95% CI) | Years | APC (95% CI) | AAPC (95% CI) |
| China | Both | 3.62 | 3.71 | 1990–2004 | 0.9 (0.5 to 1.3) | 2004–2014 | −1.6 (−2.3 to 0.8) | 2014–2019 | 2.4 (0.4 to 4.4) | 0.3 (−0.1 to 0.7) |
| | Male | 3.79 | 4.88 | 1990–1997 | −1.9 (−3.6 to 0.1) | 1997–2003 | 5.5 (2.3 to 8.8) | 2003–2019 | −0.1 (−0.6 to 0.4) | 0.6 (−0.2 to 1.4) |
| | Female | 3.44 | 2.49 | 1990–2000 | 1.8 (0.9 to 2.6) | 2000–2006 | −6.1 (−8.3 to 3.9) | 2006–2019 | −0.7 (−1.2 to 0.1) | −1.0 (−1.6 to 0.4) |
| Korea | Both | 9.59 | 3.99 | 1990–1994 | 0.0 (−4.4 to 4.5) | 1994–2019 | −3.7 (−4.0 to 3.4) | | | −3.2 (−3.8 to 2.6) |
| | Male | 8.90 | 3.32 | 1990–1995 | 1.8 (−1.6 to 5.3) | 1995–2017 | −5.0 (−5.4 to 4.7) | 2017–2019 | 2.5 (−12.0 to 19.3) | −3.4 (−4.5 to 2.2) |
| | Female | 10.29 | 4.74 | 1990–2019 | −2.7 (−2.9 to 2.5) | | | | | −2.7 (−2.9 to 2.5) |
| Japan | Both | 7.07 | 2.55 | 1990–2001 | −5.3 (−5.7 to 4.9) | 2001–2019 | −2.5 (−2.7 to 2.3) | | | −3.6 (−3.7 to 3.4) |
| | Male | 6.94 | 2.46 | 1990–2002 | −5.2 (−5.5 to 4.9) | 2002–2017 | −2.6 (−2.8 to 2.4) | 2017–2019 | 0.2 (−4.6 to 5.2) | −3.5 (−3.8 to 3.2) |
| | Female | 7.20 | 2.65 | 1990–2002 | −5.1 (−5.4 to 4.9) | 2002–2011 | −1.8 (−2.3 to 1.2) | 2011–2019 | −3.1 (−3.6 to 2.6) | −3.5 (−3.8 to 3.3) |
| USA | Both | 0.71 | 0.71 | 1990–2013 | 0.1 (−0.1 to 0.2) | 2013–2016 | 2.9 (−2.2 to 8.2) | 2016–2019 | −4.0 (−6.4 to 3.4) | −0.1 (−0.6 to 0.5) |
| | Male | 0.83 | 0.77 | 1990–2013 | −0.2 (−0.3 to 0.1) | 2013–2016 | 3.4 (−1.6 to 8.6) | 2016–2019 | −5.0 (−7.3 to 2.7) | −0.4 (−0.9 to 0.2) |
| | Female | 0.59 | 0.65 | 1990–2019 | 0.4 (0.3 to 0.5) | | | | | 0.4 (0.3 to 0.5) |

AAPC, average annual percentage change; APC, annual percentage change; ASIR, age-standardised incidence rate.

malignancies in AYA, the relative burden of GCAYA in the four countries and their changes are ranked in online supplemental figure 1. In South Korea, both in 1990 and 2019, GC was the leading burden of cancer in AYA. In China, it declined from third in 1990 to fifth in 2019. GC was once the leading cause of cancer-related DALYs in AYA in Japan and dropped to fourth in 2019. The burden of GCAYA was relatively small in the USA, ranking tenth in 1990 and then slightly rising to ninth in 2019.

### MIR of GCAYA and its changes

In 1990, the MIRs for GCAYA in China, South Korea, Japan and the USA were 0.77, 0.65, 0.38 and 0.52, respectively. From 1990 to 2019, the MIR declined constantly in South Korea, which had a higher MIR in 1990 but fell to 0.30, slightly higher than that in Japan (0.29). The MIR in China also exhibited a significant, decreasing trend, reaching 0.41 in 2019. The changing trend of MIR in the USA was not obvious; however, the MIR was 0.42 in 2019, becoming the first out of the four countries. Japan had the lowest MIR throughout the analysed period, although the decreasing trend was slight (online supplemental figure 2).

### DISCUSSION

The majority of GC occurs in elderly individuals, with its peak incidence and mortality reached among the total population aged 85–89 in China.[23] In the USA, more than 95% of GC cases are diagnosed in individuals older than 40 years.[24] Only 3.86% of new cases and 2.91% of deaths affected AYA in 2019 worldwide. GCAYA has traditionally been ignored by patients, physicians and policymakers. However, compared with older patients with GC, the burden caused by GCAYA was disproportionate, given their long life expectancy and serving as the main contributors to the economy and family care. Thus, reducing the incidence and mortality in this underserved subpopulation may benefit the development of society and the economy.

We found that nearly half of new cases and deaths of GCAYA occurred in China, which was attributed to it having the world's largest population and a higher incidence rate. The ASIR of GCAYA was much higher in the three East Asian countries, 3–5 times that in the USA. These geographic variations were also reflected in temporal trends. In Asian countries, the incidence of GCAYA showed a markedly downward trend, especially in South Korea and Japan; both had a more than 3% decrease annually. In the USA, a stable incidence was observed in males, while the ASIR in females rose steadily, although by only 0.4% per year. This is consistent with the pattern in the general population, indicating that environmental risk factors may also influence AYA, as in the elderly population.[25] In Asian countries, the high incidence of GC is closely linked to the high prevalence of *Helicobacter pylori* infection, which mainly contributes to cancers in the distal stomach.[26] In these countries, GCAYA also showed a distal predominance.[27–29] Hence, with the implementation

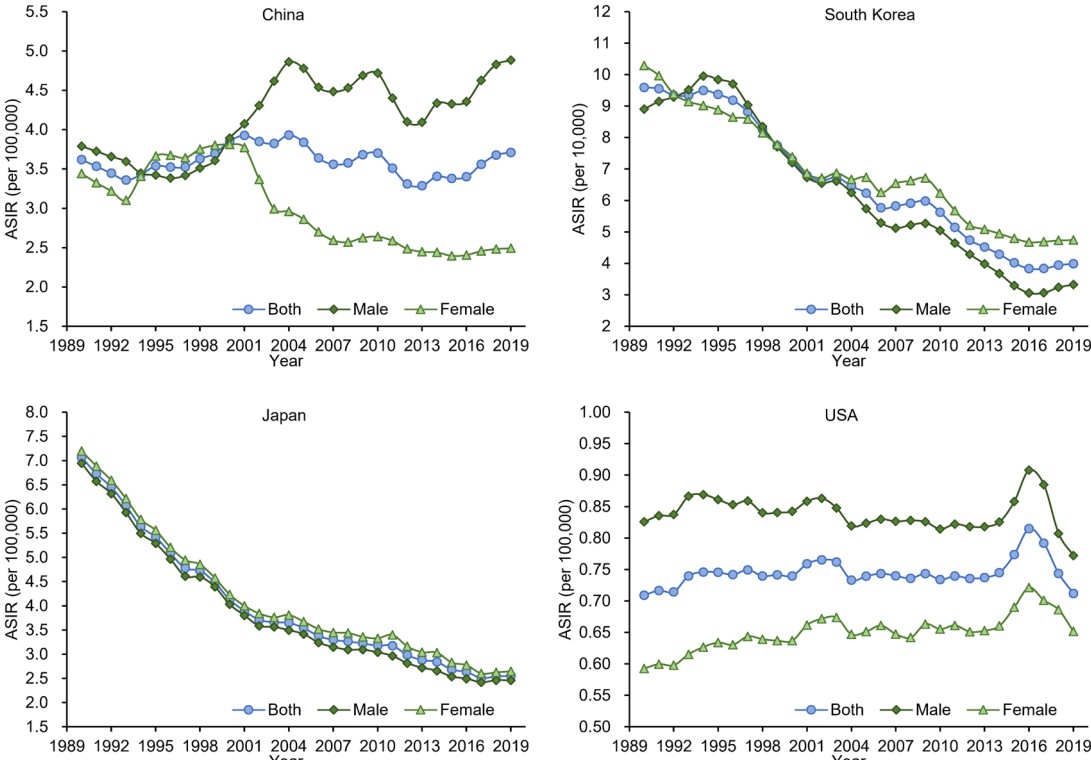

**Figure 1** The temporal trends of the age-standardised incidence rate (ASIR) for gastric cancer in adolescents and young adults by sex in China, South Korea, Japan and the USA from 1990 to 2019.

of screening and eradication programmes for this bacterium, the incidence of GC has fallen gradually, which has been called the 'epidemiology of an unplanned triumph'.[30] The effectiveness of the eradication of *H. pylori* infection to decrease the incidence of GC was also validated in many recent well-designed interventional trials.[31] Although *H. pylori* infection is primarily considered a risk factor for the development of GC in older populations, the aetiological role of *H. pylori* infection in GCAYA has also been elucidated.[32 33] Therefore, this 'unplanned triumph' has also been achieved in young adults.[34] In addition, modern practices of food preservation and refrigeration have increased the consumption of fresh fruits and vegetables, which are protective factors for GC.[35] In contrast, the risk factors associated with GC in the USA were somewhat different from those in Asian countries. Some authors have suggested that increased salt intake and obesity may contribute to an increased incidence of GCAYA.[6 36] These risk factors are mainly associated with proximal GC, which cannot be distinguished in this study; however, the increasing trend in GCAYA is consistent with the dramatic shift in the location of GC that has occurred in the USA, with a marked increase in diffuse-type GC of the proximal stomach.[24 37 38]

In addition to the differences in risk factors, different forms of screening and early detection programmes among the four countries may explain the variations in incidence and its time trends. As early as the 1960s, Japan began to implement a mass GC screening, which was expanded for all residents older than 40 years in 1983.[7] In South Korea, GC screening started in 1999 and expanded nationwide in

2002.[8] GC screening programmes were launched much later in China, and the objects were limited to selected individuals with high-risk factors.[8] In contrast, to date, there have been no nationwide GC screening programmes in the USA. The effects of these programmes on the incidence of GC are contradictory, and recently published well-designed studies have shown that screening programmes effectively decrease the GC incidence.[39 40] Although these programmes did not cover the AYA populations, the changing trends of the ASIR of GCAYA in the four countries may partially reflect the effects of these programmes. Because of the early establishment of GC screening and early diagnosis programmes, the incidence of GCAYA decreased steadily in South Korea and Japan during the analysis period. In China, the change among the entire period was not apparent, which may have resulted from the first increase after the implementation of screening programmes, which in turn might detect more new cases. Next, the incidence began to decline due to the effects of these programmes. How GC screening programmes can decrease the incidence of GC is not clear, especially in AYA, which was not covered by these programmes. This could be explained by the fact that the implementation of GC screening programmes may increase the awareness of GC in the entire population. This would also encourage young people to undergo GC-specific examinations. *H. pylori* infection can be diagnosed by these examinations, leading to the eradication of this bacterium and a decrease in *H. pylori*-related GCs. Furthermore, electronic endoscopy has been widely accepted as the first method for GC screening, which may detect more precancerous benign

**Table 3** The temporal trend in the mortality rate of gastric cancer in adolescents and young adults from 1990 to 2019 in China, South Korea, Japan and the USA

| Country | Sex | ASMR (per 100 000) | | Trends 1 | | Trends 2 | | Trends 3 | | 1990–2019 |
|---|---|---|---|---|---|---|---|---|---|---|
| | | 1990 | 2019 | Years | APC (95% CI) | Years | APC (95% CI) | Years | APC (95% CI) | AAPC (95% CI) |
| China | Both | 2.80 | 1.50 | 1990–2003 | 0.0 (−0.4 to 0.3) | 2003–2013 | −5.1 (−5.7 to 4.4) | 2013–2019 | −0.8 (−2.0 to 0.5) | −2.0 (−2.3 to 1.6) |
| | Male | 2.90 | 1.91 | 1990–1997 | −2.7 (−4.8 to 0.5) | 1997–2003 | 3.8 (0.0 to 7.8) | 2003–2019 | −3.5 (−4.1 to 2.9) | −1.8 (−2.7 to 0.9) |
| | Female | 2.69 | 1.07 | 1990–2000 | 0.8 (0.0 to 1.6) | 2000–2007 | −7.9 (−9.5 to 6.3) | 2007–2019 | −3.3 (−3.9 to 2.8) | −3.1 (−3.6 to 2.6) |
| Korea | Both | 6.29 | 1.18 | 1990–1995 | −4.6 (−6.7 to 2.4) | 1995–2016 | −6.8 (−7.0 to 6.5) | 2016–2019 | 0.9 (−4.0 to 6.1) | −5.6 (−6.2 to 5.0) |
| | Male | 5.66 | 0.95 | 1990–1994 | −1.1 (−5.1 to 3.0) | 1994–2016 | −7.8 (−8.1 to 7.5) | 2016–2019 | 1.9 (−4.5 to 8.7) | −6.0 (−6.8 to 5.2) |
| | Female | 6.94 | 1.44 | 1990–2016 | −5.8 (−6.0 to 5.6) | 2016–2019 | 0.5 (−4.3 to 5.5) | | | −5.2 (−5.7 to 4.7) |
| Japan | Both | 2.69 | 0.73 | 1990–2003 | −5.6 (−5.8 to 5.4) | 2003–2017 | −3.8 (−4.0 to 3.6) | 2017–2019 | −0.0 (−3.9 to 3.9) | −4.4 (−4.7 to 4.1) |
| | Male | 2.30 | 0.69 | 1990–2003 | −5.2 (−5.5 to 5.0) | 2003–2017 | −3.6 (−3.8 to 3.4) | 2017–2019 | 1.0 (−3.3 to 5.5) | −4.0 (−4.3 to 3.7) |
| | Female | 3.08 | 0.77 | 1990–2003 | −5.9 (−6.1 to 5.7) | 2003–2017 | −4.1 (−4.3 to 3.9) | 2017–2019 | −0.6 (−4.6 to 3.6) | −4.7 (−4.9 to 4.4) |
| USA | Both | 0.37 | 0.30 | 1990–2013 | −0.8 (−0.9 to 0.7) | 2013–2016 | 3.6 (0.3 to 6.9) | 2016–2019 | −3.6 (−5.2 to 2.0) | −0.7 (−1.0 to 0.3) |
| | Male | 0.41 | 0.30 | 1990–2013 | −1.2 (−1.2 to 1.1) | 2013–2016 | 4.2 (0.0 to 8.7) | 2016–2019 | −5.0 (−7.0 to 3.0) | −1.0 (−1.5 to 0.6) |
| | Female | 0.33 | 0.29 | 1990–2013 | −0.4 (−0.5 to 0.3) | 2013–2016 | 2.9 (−1.6 to 7.6) | 2016–2019 | −2.7 (−4.8 to 0.5) | −0.3 (−0.8 to 0.2) |

AAPC, average annual percentage change; APC, annual percentage change; ASMR, age-standardised mortality rate.

lesions or in situ neoplasms. Thus, in the USA without GC screening programmes, the incidence of GCAYA showed a stable trend in both sexes combined and increased steadily in females at 0.4% annually.

With regard to the mortality of GCAYA, regardless of deaths or ASMR, both showed significant downward trends among the four countries. The changing patterns in mortality reflect shifting patterns not only in terms of incidence but also in case fatalities, which we represented with MIR in this study.[13] Thus, a great decline in mortality was observed in Japan and South Korea, in which there was an impressive decrease in incidence and MIR. Case fatality (MIR) was determined primarily by advancements in therapy and early detection. Under the current concept of multidisciplinary therapy for GC, modern treatment methods have significantly increased the cure rate of localised GC and prolonged the survival of advanced GC.[41] However, in this study, we found that the MIR in the USA in 1990 was lower than that of China and South Korea, but it ranked first among the four countries in 2019, despite its highly developed healthcare system. This may have stemmed from the advanced stages of GCAYA diagnosed in the USA, increasing incidence in females, and the striking health disparities observed in cancers,[42] which balanced the improvement of therapy strategies. In Japan, the MIR of GCAYA was continuously the lowest during the analysis period, while in South Korea, it was gradually close to that of Japan starting in 2008. This phenomenon indicates that the most effective strategy to decrease the mortality of GCAYA is screening and early diagnosis. Therefore, according to recent studies, the prevalence of early GC rose from 28.6% in 1995 to 58.0% in 2007 in South Korea, and a 57% GC mortality rate reduction was attributed to endoscopic screening in Japan.[43 44]

Despite the decline in incidence and mortality of GCAYA in South Korea and Japan throughout the analysis period, the mortality tended to be stable in 2016. This implies that the effects of current prevention and screening programmes for GC have reached their limitations in AYA. In addition, distinctive etiological characteristics have been recognised in GCAYA. Approximately 10% of GC cases showed familial clustering, which was more notable in GCAYA.[45 46] Up to 3% of GC cases are related to inherited cancer predisposition syndromes, including hereditary diffuse gastric cancer (HDGC), familial adenomatous polyposis and Lynch syndrome, all of which predispose younger populations to GC development.[47 48] HDGC is an autosomal dominant syndrome arising from germline mutations in the tumour suppressor gene CDH1 and is characterised by the development of GCs, predominantly the diffuse type and occurs in females at a young age.[47 49] These characteristics are consistent with diffuse GC and female predominance, reflecting the hereditary factors may contribute to the carcinogenesis of GCAYA. These hereditary factors are irreversible with current technological capabilities, and the best way to decrease the deaths caused by GC in these patients is precursor lesion detection by endoscopic surveillance and prophylactic total gastrectomy.[47 50] However, these specific cancer types still account for a minority of the total burdens

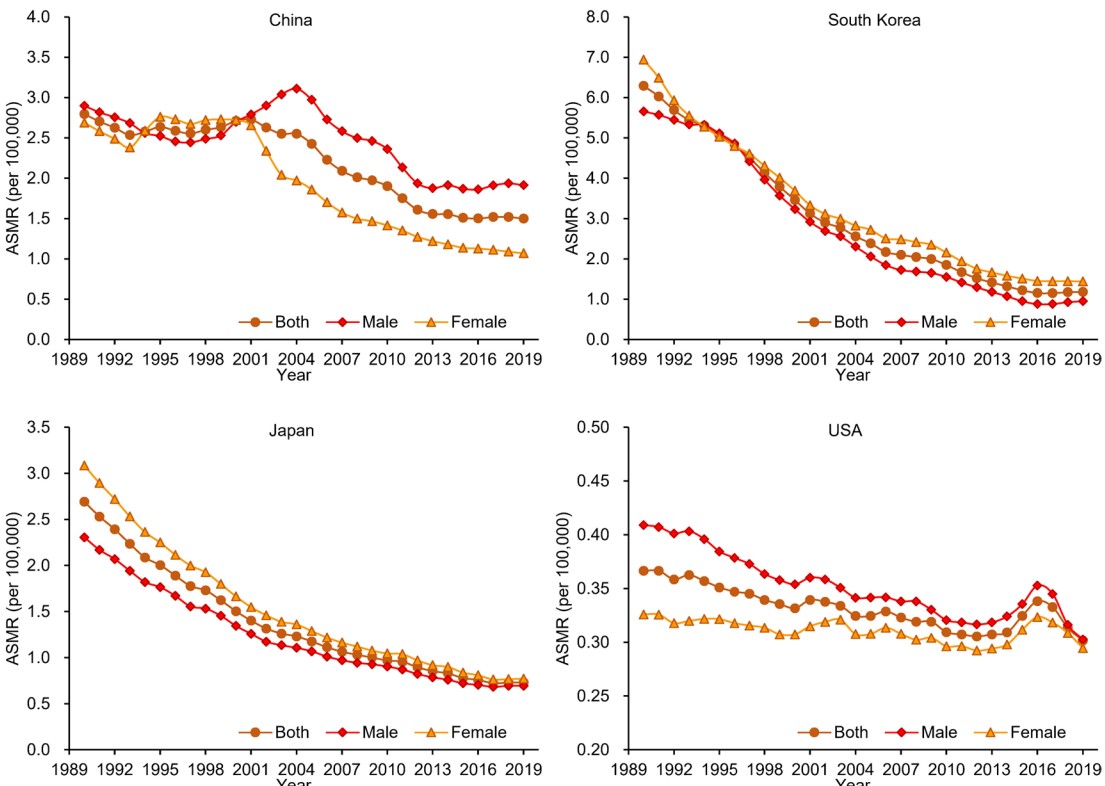

**Figure 2** The temporal trends of the age-standardised mortality rate (ASMR) for gastric cancer in adolescents and young adults by sex in China, South Korea, Japan and the USA from 1990 to 2019.

caused by GCAYA. Other relevant opportunities to further improve the outcomes of GCAYA are worthwhile. Because the incidence of GC was low in AYA, endoscopic screening was considered to be associated with a low yield rate and not cost-effective.[51] However, the burdens caused by GC are not small in AYA. Despite the significant decrease, GC still ranked first, fourth and fifth among all cancer types in AYA in South Korea, Japan and China, respectively, with regard to DALYs. Although it was relatively small, the burden caused by GCAYA in the USA increased from tenth in 1990 to ninth in 2019. In addition, as mentioned earlier, the AYA population has a long life expectancy and contributes greatly to society and the economy. Hence, prevention and screening among AYA in regions with a higher incidence of GC is worthwhile, and research into screening programmes specifically in AYA is needed to determine the benefits and potential risks.

Our findings allow for a comprehensive estimation and comparison of the GCAYA burden among China, South Korea, Japan and the USA; however, several limitations exist, which were also described in studies using data from GBD 2019 and in studies on cancer incidence in AYA.[10 15 17] First, although GBD 2019 used many strategies to improve the data quality and comparability, they were obtained from selected registries and might not be accurate in reflecting the overall burden in some countries, particularly for countries where data are not available or are of poor quality, which may affect the integrity and accuracy of the data that we analysed. Second, we were unable to analyse cardia and non-cardia GC separately, two subtypes that have different risk factors and temporal incidence trends.[52 53] Third, the incidence

and mortality were low and volatile, especially in the USA, which means that even the smallest change could lead to a significant analytical outcome, especially when determined with a very short duration. Despite these limitations, our study involved data retrieved from the GBD 2019, the best data currently available for a long time period. Our findings highlight the health burden of GCAYA and the effects of prevention and screening programmes among GCAYA, as well as the need to increase awareness and resources for this neglected subpopulation.

Overall, we have offered a comprehensive analysis and comparison of the burden and temporal trends of GCAYA in China, South Korea, Japan and the USA. In the past three decades, the incidence and mortality of GCAYA have been declining significantly in South Korea and Japan. A falling trend also appeared for females in China in recent years, while a steadily slowly rising trend has been observed for females in the USA. Although not covered by prevention and screening programmes, these variations in incidence and mortality of GCAYA may reflect variations in risk factors, cancer control strategies and treatment accessibility of GC among the four countries. Although GC is much less frequently diagnosed in AYA than in older populations, its effects remain considerable due to the long life expectancy of these individuals. Investigating the reasons behind the varying disease burden and changing trends of GCAYA across countries will inform recommendations for prevention measures and timely diagnosis specific to this underserved population to further decrease the GC burden.

**Contributors** Conceptualisation: JL and JSW. Data curation: SLW, YZ and YF. Formal analysis: SLW, YZ, YF and JL. Methodology: SLW, JSW and JL. Software: JL. Supervision: JSW and JL. Roles/writing-original draft: all authors. JL is responsible for the overall content as the guarantor.

**Funding** This study was supported by Scientific Research Projects of Health Commission of Mianyang City (202012).

**Disclaimer** The funders had no role in the design and conduct of the study; collection, management, analysis and interpretation of the data; preparation, review or approval of the manuscript; and decision to submit the manuscript for publication.

**Competing interests** None declared.

**Patient and public involvement** Patients and/or the public were not involved in the design, or conduct, or reporting, or dissemination plans of this research.

**Patient consent for publication** Not applicable.

**Ethics approval** This study was approved by the Academic Committee of the Third Hospital of Mianyang (20190307).

**Provenance and peer review** Not commissioned; externally peer reviewed.

**Data availability statement** Data are available in a public, open access repository.

**ORCID iD**
Jian Li http://orcid.org/0000-0002-5807-2360

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
