## [Reviewer comments · BMJ Open]

ARTICLE DETAILS

TITLE (PROVISIONAL)	Gastric cancer incidence, mortality, and burden in adolescents and young adults: A time-trend analysis and comparison among China, South Korea, Japan and the USA
AUTHORS	Wu, Si; Zhang, Yao; Fu, Yi; Li, Jian; Wang, Ji

VERSION 1 – REVIEW

REVIEWER	Bornschein, Jan Oxford University Hospitals NHS Foundation Trust, Translational Gastroenterology Unit
REVIEW RETURNED	03-Mar-2022

GENERAL COMMENTS	Thank you for the opportunity to review this interesting paper by Wu et al on the epidemiological trends of gastric cancer in the young population of China, South Korea, Japan and the USA. While it is very important to gather more information on the disease pattern in this population, there are several issues with regards to this article that I think need further input. Major: - Abstract: The last sentence of the results is quite striking but not necessarily supported to that degree by the data presented. This is mostly due to the limitation of the study/data listed by the authors themselves. I would be grateful if this could be rephrased to take the emphasis off this item.- The distinction between junctional and non-junctional gastric cancers (I would be grateful if the terms 'proximal' and 'distal gastric cancer' could be avoided – especially when talking about H. pylori) is of high relevance since we know that there is a different distribution in Eastern and Western population due to different risk factor exposure. This needs more careful discussion and weighing up in view of the conclusions.- Results: I found it quite difficult to read the results paragraphs since they seem to list a lot of data in a slightly confusing order with sometimes random facts being listed. I would be grateful if this could be revised and more structured alongside the aims and objectives that should be stated clearly in the introduction.- Results/Tables: You mention 5-year age brackets selected for analyses, but these don't appear anywhere in the results. I would be quite keen on seeing the data reflecting different age groups in your cohort and pointing out relevant differences here.- Results/Tables: Both in the text and in the tables seemingly random timeframes are reported. While these might reflect certain trends in the incidence and mortality curves for each country, it makes comparison between countries a bit difficult and the whole flow of the paper more difficult to follow. I would stick to pre-
---

	selected periods and add additional interpretation and comments in the discussion.  - Discussion: It is important that the authors stick strictly to the correct definitions. While some aspects are mentioned in the paper, the context is not always clear and the discussion in particular needs a bit of restructuring. There is a difference between the screen-and-treat programs for H. pylori which have a preventive impact on GC incidence in the respective countries and the screening programs for gastric cancer. The latter help with early detection but won't have the same impact on GC incidence. Thus, gastric cancer screening efforts will rather improve mortality – due to the early detection – but less so incidence. This needs to be clearly distinguished and addressed, especially when comparing different countries and different timeframes. The fact that early detection efforts don't cover the younger population needs also addressing and needs to be strictly reflected when interpreting the data from different countries. Some of the conclusions are rather conjecture. - Discussion: The section on the impact of familial/hereditary cancers needs more emphasis. This affects usually more the younger population, shows less dominance of male sex and is usually resulting in cancers of the diffuse type. (At no point the paper is there any comment on the distinction between intestinal and diffuse type cancers.) Minor:  - Introduction: The authors point out several facts regarding the background information available on the topic, but the introduction section does not deliver a clear focus on the main aims of the study and the rationale for the different calculations. - Methods: Why was the timeframe defined as 1990-2010 when you analysed data up to 2019? - Results: Page 9, line 9: 'As of the writing of this paper' – could you please define a precise time point (e.g. November 2021 or similar). - Discussion, page 10, line 28: I would be grateful if the phrase could be revised that states 'GCAYA has traditionally been ignored'. - Discussion: The comments on the socio-economic impact of GCAYA is very interesting and worth considering, but are there any data to support the suggestions made here? - Discussion: The paper referring to an 'unplanned triumph' was published in 1986 when the main impact of H. pylori eradication on GC prevention was not yet that obvious. Maybe the authors can elaborate a bit more on this issue. - Discussion: The authors mention the 'highly developed health care system' in the USA. While it is true that technically advanced diagnostic and treatment options are available these are often not accessible to vast parts of the general population due to a lack of insurance cover etc. This needs to be reflected. - Discussion: The data on MIR is very important and needs more reflection. - Discussion: Page 14, line 47. 'Bias is inevitable' – which bias? Please be more precise. - Discussion: the discussion in general needs a clearer focus regarding the intention of this study and the take home messages.
REVIEWER	Tavares, Amélia Centro Hospitalar de Vila Nova de Gaia

REVIEW RETURNED	24-Apr-2022
-------------

GENERAL COMMENTS	This is an article of public and current interest. However, in certain parts it is very difficult to read, especially the results, with recurrent use of non-simple abbreviations, making it difficult to follow the results. The authors do not explain why they decided to select these 4 countries to assess the incidence and mortality of gastric cancer in young patients, that is, was it due to the incidence and prevalence of this pathology in these countries? The introduction should provide a better contextualization about gastric cancer. The variables studied must be better explained, that is, in clinical terms what they mean and what they imply. In the methods, page 5, line 33, "1990-2010" is mentioned, but the study refers to the period from 1990 to 2019. The results must be expressed in a simpler and more direct way. On page 6, line 51, values that are not shown in table 1 are mentioned. In the results the abbreviations are very frequently used and sometimes it is difficult to know what each abbreviation means, there should be greater care in this reference. The discussion is adequate and correct, however the authors should improve the conclusions. Regarding references, they use the data from GLOBOCAN 2018 in the first reference, when more current data from 2020 are already available.
---

VERSION 1 – AUTHOR RESPONSE

Reviewer: 1

Thank you for the opportunity to review this interesting paper by Wu et al on the epidemiological trends of gastric cancer in in the young population of China, South Korea, Japan and the USA. While it is very important to gather more information on the disease pattern in this population, there are several issues with regards to this article that I think need further input.

Major:

- Abstract: The last sentence of the results is quite striking but not necessarily supported to that degree by the data presented. This is mostly due to the limitation of the study/data listed by the authors themselves. I would be grateful if this could be rephrased to take the emphasis off this item.

Response:

Thank you for your comments. We agree with the reviewer that due to the limitations of the study/data, the constant decline in MIR cannot be concluded in the USA; however, the changing trends are obvious in China and South Korea. We revised the results in the abstract and main text to emphasize this item.

- The distinction between junctional and non-junctional gastric cancers (I would be grateful if the terms 'proximal' and 'distal gastric cancer' could be avoided – especially when talking about H. pylori) is of high relevance since we know that there is a different distribution in Eastern and Western population due to different risk factor exposure. This needs more careful discussion and weighing up in view of the conclusions.

Response:

According to your comments, we have added some more discussion about the risk factors and epidemiological characteristics of proximal and distal GC, which may possibly underlie some results found in our study. However, although the proximal and distal GC classification is not stringent and the GEJ and non-GEJ GC classification has popularized in recent years, as the majority of articles

cited in this manuscript divided GC into proximal and distal cancers, for not mis-citing, we also adopted the proximal and distal GC classification.

- Results: I found it quite difficult to read the results paragraphs since they seem to list a lot of data in a slightly confusing order with sometimes random facts being listed. I would be grateful if this could be revised and more structured alongside the aims and objectives that should be stated clearly in the introduction.

Response:

Thank you for your comments. We agree with the reviewer that the results section in its previous edition may be somewhat confusing for readers who are not familiar with the distribution pattern of the data; therefore, we revised this section in two ways. On the one hand, the 95% CI values of new cases, deaths and DALYs were deleted, which may not interfere with the interpretation of the data; on the other hand, this section was re-expressed in a much simpler and direct way. In addition, we have added more explanations to the introduction as to why we chose the four countries in this study.

- Results/Tables: You mention 5-year age brackets selected for analyses, but these don't appear anywhere in the results. I would be quite keen on seeing the data reflecting different age groups in your cohort and pointing out relevant differences here.

Response:

In this study, the data were collected in a 5-year age bracket for the calculation of age-standardized rates in adolescents and young adults, which was not available in GBD 2019. Therefore, the data are not listed in the manuscript. Although variations may exist among these different age groups, as numerous data will be added to further lower the readability of this manuscript, we hope you will agree with us not to include this analysis in this manuscript, which will not limit the aim of our study.

- Results/Tables: Both in the text and in the tables seemingly random timeframes are reported. While these might reflect certain trends in the incidence and mortality curves for each country, it makes comparison between countries a bit difficult and the whole flow of the paper more difficult to follow. I would stick to pre-selected periods and add additional interpretation and comments in the discussion.

Response:

Thank you for your suggestion. The time trend analysis for ASIR and ASMR was performed by Joinpoint software. When the point number was set, the time period was determined by the software, which was chosen to fit the best model. In addition, our primary aim is to discover the differences in changing trends (including the magnitude, direction and time period) among the four countries and to inform the possible explanations that may help future cancer control strategy establishment.

- Discussion: It is important that the authors stick strictly to the correct definitions. While some aspects are mentioned in the paper, the context is not always clear and the discussion in particular needs a bit of restructuring. There is a difference between the screen-and-treat programs for H. pylori which have a preventive impact on GC incidence in the respective countries and the screening programs for gastric cancer. The latter help with early detection but won't have the same impact on GC incidence. Thus, gastric cancer screening efforts will rather improve mortality – due to the early detection – but less so incidence. This needs to be clearly distinguished and addressed, especially when comparing different countries and different timeframes. The fact that early detection efforts don't cover the younger population needs also addressing and needs to be strictly reflected when interpreting the data from different countries. Some of the conclusions are rather conjecture.

Response:

Thank you for your comments. However, we do not entirely agree with you on this point. Although the impacts and the underlying mechanisms of screening strategies on GC incidence are inconclusive, many studies support that screening can decrease the incidence of GC, especially two well-designed studies published recently. We have added them to the discussion. In addition, as mentioned above, due to the relatively low incidence of GC in populations, it is difficult and even infeasible to study the associated issues of GC in a local region. Therefore, we tried to analyze high-quality data obtained from large datasets such as GBD 2019. Although such results are inevitably correlative rather than causative, as many propositions can be raised from the results of epidemiological studies and validated to be true or not, they may still inform some important conceptual and practical changes

to the management of GCAYA, which cannot be provided by few cohort studies attributed to the low incidence.

- Discussion: The section on the impact of familial/hereditary cancers needs more emphasis. This affects usually more the younger population, shows less dominance of male sex and is usually resulting in cancers of the diffuse type. (At no point the paper is there any comment on the distinction between intestinal and diffuse type cancers.)

Response:

Thank you for your suggestion. We have discussed more detailedly the hereditary GC with its diffuse type and female predominance and its association with GCAYA.

Minor:

- Introduction: The authors point out several facts regarding the background information available on the topic, but the introduction section does not deliver a clear focus on the main aims of the study and the rationale for the different calculations.

Response:

We have added more explanations to the introduction as to why we chose the four countries in this study, which also indicated the aim of our study. In addition, as a secondary analysis design, this study only describes the calculation methods of parameters such as AAPC, while other data collection and processing methods are described by the GBD 2019 collaborator elsewhere; therefore, we only refer the readers to these articles.

- Methods: Why was the timeframe defined as 1990-2010 when you analysed data up to 2019?

Response:

Thank you for your comments. It is a clerical error and was corrected.

- Results: Page9, line 9: 'As of the writing of this paper' – could you please define a precise time point (e.g. November 2021 or similar).

Response:

Response:

We apologize for our imprecise expression, and we have revised it to "until 2019".

- Discussion, page 10, line 28: I would be grateful if the phrase could be revised that states 'GCAYA has traditionally been ignored'.

Response:

It has been revised as the comment.

- Discussion: The comments on the socio-economic impact of GCAYA is very interesting and worth considering, but are there any data to support the suggestions made here?

Response:

We did not find any specific data to support that suggestion, although many groups, including AYAO PRG, have made similar statements. However, it may be proposed as a common sense. Young adults represent the main proportion of contributors to the economy and their family care; therefore, we think a GC afflicting young adults may cause more socioeconomic influence than an elderly patient.

- Discussion: The paper referring to an 'unplanned triumph' was published in 1986 when the main impact of H. pylori eradication on GC prevention was not yet that obvious. Maybe the authors can elaborate a bit more on this issue.

Response:

Thank you for your comments. We agree that when the notion 'unplanned triumph' was proposed, the main impact of H. pylori eradication on GC prevention was not yet obvious. Therefore, we cited a review that summarizes the evidence to support the effectiveness of the eradication of H. pylori infection to decrease the incidence of GC.

- Discussion: The authors mention the 'highly developed health care system' in the USA. While it is true that technically advanced diagnostic and treatment options are available these are often not accessible to vast parts of the general population due to a lack of insurance cover etc. This needs to be reflected.

Response:

Thank you for your comments. The review by Alcaraz KI et al. comprehensively discussed the social determinants to advance cancer health equity in the United States. We cited this review in our discussion, which may also reflect the distinct MIR manifestation, as you have commented.

- Discussion: The data on MIR is very important and needs more reflection.

Response:

We have slightly revised the results section on the MIR data. However, in the discussion section, a paragraph has been allocated to discuss this topic, and we have not found appropriate content to be revised.

- Discussion: Page 14, line 47. 'Bias is inevitable' – which bias? Please be more precise.

Response:

Thank you for your suggestion. We have revised this as “they were obtained from selected registries and might not be accurate to reflect the overall burden in some countries, particularly for countries where data are not available or are of poor quality”.

- Discussion: the discussion in general needs a clearer focus regarding the intention of this study and the take home messages.

Response:

Many parts of the discussion have been revised according to your constructive comments, which may make it clearer and more focused. However, when we re-examined the discussion again, we found that it was also too long for a reader, which may be a common limitation of such a study type based on so many parameters. For readers familiar with such study design, this structure does not confuse their interpretation.

Reviewer: 2

This is an article of public and current interest.

However, in certain parts it is very difficult to read, especially the results, with recurrent use of non-simple abbreviations, making it difficult to follow the results. The authors do not explain why they decided to select these 4 countries to assess the incidence and mortality of gastric cancer in young patients, that is, was it due to the incidence and prevalence of this pathology in these countries? The introduction should provide a better contextualization about gastric cancer.

Response:

Thank you for your constructive comments. As mentioned above, because of the relatively low incidence of GCAYA in populations, it is difficult and even infeasible to study the associated issues of GCAYA in a local region. Therefore, we tried to analyze high-quality data obtained from the large GBD 2019 dataset. Furthermore, we have added more explanations to the introduction as to why we chose the four countries in this study. Because the risk factors, incidence, cancer control strategies and health-care system varied among the selected countries, we hope that the various disease burden parameters and changing trends among them may propose some conceptual and practice improvements relevant to the management of GCAYA. In addition, with regard to your concerns about abbreviations, we reexamined our manuscript and found that except for GCAYA, all several other abbreviations are commonly used in the literature, even for gastric cancer in adolescents and young adults. Many authors have adopted GCAYA to avoid repeated long descriptions for easy reading. Therefore, we hope you can agree with the retaining of these abbreviations.

The variables studied must be better explained, that is, in clinical terms what they mean and what they imply.

Response:

We have added a “definition” section, mainly aimed to explain the definition of GCAYA and DALY. The MIR was explained in the “statistical analysis” section, while other detailed estimation methods for incidence, mortality, and DALYs were referred to relevant references.

In the methods, page 5, line 33, "1990-2010" is mentioned, but the study refers to the period from 1990 to 2019.

Response:

Thank you for your comments. It is a clerical error and was corrected.
The results must be expressed in a simpler and more direct way.

Response:

Thank you for your comments. We agree with the reviewer that the results section in its previous edition may be somewhat confusing for readers who are not familiar with the distribution pattern of the data; therefore, we revised this section in two ways. On the one hand, the 95% CI values of new cases, deaths and DALYs were deleted, which may not interfere with the interpretation of the data; on the other hand, this section was re-expressed in a much simpler and direct way.

On page 6, line 51, values that are not shown in table 1 are mentioned.

Response:

We have revised the description.

In the results the abbreviations are very frequently used and sometimes it is difficult to know what each abbreviation means, there should be greater care in this reference.

Response:

The comment has been responded as above.

The discussion is adequate and correct, however the authors should improve the conclusions.

Response:

Thank you for your comments. We have improved the expression of the conclusions to avoid overextrapolating the results.

Regarding references, they use the data from GLOBOCAN 2018 in the first reference, when more current data from 2020 are already available.

Response:

Thank you for your comments. We have updated the reference.

VERSION 2 – REVIEW

REVIEWER	Bornschein, Jan Oxford University Hospitals NHS Foundation Trust, Translational Gastroenterology Unit
REVIEW RETURNED	04-Jul-2022

GENERAL COMMENTS	Thank you for revising your paper and for your detailed response to my comments. I appreciate the effort spent and think that the paper has indeed improved. There are a few issues on which we don't share the same opinion (e.g. nomenclature regarding subtypes of gastric cancer, impact of screening on GC incidence), but I accept the authors' viewpoint on this. There are only two minor point I would like to raise: - H. pylori should always be put in Italics. - I would still appreciate a comment on the issue of proximal/distal vs junctional/non-junctional gastric cancer, even with the paper sticking to the old-fashioned classification (see also previous comments and authors' response). however, I am happy to leave this to the editors if this is considered necessary.
--

VERSION 2 – AUTHOR RESPONSE

Reviewer: 1

Thank you for revising your paper and for your detailed response to my comments. I appreciate the effort spent and think that the paper has indeed improved. There are a few issues on which we don't share the same opinion (e.g. nomenclature regarding subtypes of gastric cancer, impact of screening

on GC incidence), but I accept the authors' viewpoint on this. There are only two minor point I would like to raise:

- *H. pylori* should always be put in Italics.

- I would still appreciate a comment on the issue of proximal/distal vs junctional/non-junctional gastric cancer, even with the paper sticking to the old-fashioned classification (see also previous comments and authors' response). however, I am happy to leave this to the editors if this is considered necessary.

Response:

Thank you for your understanding. With regard to the nomenclature of subtypes of gastric cancer based on location, we indeed prefer the junctional/non-junctional classification, which popularized in recent years, however, as we explained in previous responses, we stick to the old-fashioned classification in this manuscript. This also indicates the necessity to reach a consensus on such topics, which may benefit comparison between studies and management of gastric cancer in practice.

In addition, thank you for pointing out our mistake, we have revised them in Italics.